# Low-Intensity Pulsed Ultrasound Counteracts Advanced Glycation End Products-Induced Corpus Cavernosal Endothelial Cell Dysfunction via Activating Mitophagy

**DOI:** 10.3390/ijms232314887

**Published:** 2022-11-28

**Authors:** Yuzhuo Chen, Ming Xiao, Liangyu Zhao, Yongquan Huang, Yuhong Lin, Ting Xie, Jiali Tian, Qi Wang, Yuxin Tang, Zhongzhen Su

**Affiliations:** 1Department of Ultrasound, The Fifth Affiliated Hospital, Sun Yat-sen University, Zhuhai 519000, China; 2Guangdong Provincial Key Laboratory of Biomedical Imaging, Department of Interventional Medicine, The Fifth Affiliated Hospital, Sun Yat-sen University, Zhuhai 519000, China; 3Department of Urology, The Fifth Affiliated Hospital, Sun Yat-sen University, Zhuhai 519000, China

**Keywords:** low-intensity pulsed ultrasound (LIPUS), regenerative medicine, diabetes mellitus-induced erectile dysfunction (DMED), corpus cavernosal endothelial cells (CCECs), angiogenesis, mitophagy

## Abstract

Injury to corpus cavernosal endothelial cells (CCECs) is an important pathological basis of diabetes mellitus-induced erectile dysfunction (DMED), while low-intensity pulsed ultrasound (LIPUS) has been shown to improve erectile function in DMED. To further understand its therapeutic mechanism of action, in this study, we first demonstrated increased apoptosis and shedding in the CCECs of DMED patients, accompanied by significant mitochondrial injury by immunohistochemistry and electron microscopy of corpus cavernosum tissue. Next, we used advanced glycation end products (AGEs) to simulate the diabetic environment in vitro and found that AGES damaged mitochondria and inhibited angiogenesis in CCECs in a dose-dependent manner, while LIPUS treatment significantly reversed its effects. Mechanistic studies based on transcriptome sequencing showed that LIPUS significantly up-regulated LC3 and PARKIN protein levels in mitochondria, promoted mitophagy, and affected mitochondrial dynamics and reactive oxygen species (ROS) production. In addition, the protective effects of LIPUS were abrogated when mitophagy was inhibited by 3-methyladenine. In summary, LIPUS exerted potent inhibitory effects on AGES-induced CCEC failure via mitophagy, providing a theoretical basis for DMED treatment that encompasses the protection of endothelial structure and function.

## 1. Introduction

Diabetes mellitus-induced erectile dysfunction (DMED) is a common complication of diabetes that affects more than half of male diabetes patients, with a prevalence odd of approximately 3.5 times more than healthy men [1]. One of the most important cellular processes underlying pathological damage in diabetes is the increased formation of advanced glycation end products (AGEs). Proteins are modified into harmful AGEs with altered functions when their amino acid constituent groups interact with AGE precursors such as glyoxal, methylglyoxal, and deoxyglucosone [2].

Although the etiology of DMED is multifactorial, endothelial dysfunction is recognized as a key factor in the pathophysiology of the disease [3]. AGEs play an important role in diabetic vasculopathy, and the levels of AGEs are correlated with the severity of diabetic vascular complications [4]. The diabetic state perturbs vascular endothelial function by disrupting the generation and release of vasoactive substances and inhibiting endothelium-derived hyperpolarization [5,6]. Furthermore, AGEs damage vascular structure via promoting apoptosis, calcification, senescence, and hyperpermeability in endothelial cells (ECs) [7,8]. Erectile dysfunction (ED) is known as a predictor of cardiovascular disease events, and we have shown that corpus cavernosum endothelial cells (CCECs) are more susceptible to injury than the ECs of other blood vessels [9,10]. However, common and unique factors relating to the damage of CCECs and other types of ECs in diabetes remain unclear.

Once formed, AGEs can bind to different AGE receptors to promote the generation of reactive oxygen species (ROS), invariably favoring oxidative stress [2]. In diabetic conditions, mitochondria produce excessive superoxide anion radicals, which overload the antioxidant systems and induce oxidative stress, inflicting damage on nuclear DNA as well as other biomolecules [11,12]. Furthermore, cellular oxidative stress caused by diabetes can exacerbate mitochondrial dysfunction, creating a vicious cycle [12]. Mitophagy, a type of autophagy that takes place in dysfunctional mitochondria, plays a key role in mitochondrial quality control. Recent studies have suggested that mitophagy plays a protective role in diabetic vascular complications, principally through the clearance of abnormal mitochondria, thereby preventing oxidative stress and reducing cell apoptosis [13,14]. Although the role of mitophagy in the pathogenesis of DMED has not been reported, the similar histological features of the penile cavernosa and blood vessels in the diabetic state suggest that mitophagy may be a potential therapeutic target.

Low-intensity pulsed ultrasound (LIPUS) is a specific type of ultrasound that delivers at a low intensity and outputs in the mode of pulsed waves. It produces minimal thermal effects while maintaining the transmission of acoustic energy to the target tissue, providing noninvasive physical stimulation for therapeutic applications [15]. The use of LIPUS has been demonstrated to have beneficial effects on bone healing, soft-tissue regeneration, angiogenesis, neuromodulation, and inflammation via mechanical stress effects and other mechanisms [15]. In addition, LIPUS therapeutic devices have been shown to improve erectile function non-invasively in ED patients and diabetes-induced animal models of ED [16]. Its therapeutic mechanisms of action on endothelial function may include effects on cell proliferation and differentiation, extracellular matrix remodeling, permeabilization of biological barriers, and oxidative stability [17,18,19]. Recently, Tian Guo et al. reported that LIPUS therapy inhibited b-cell dysfunction induced by high lipid levels via effects on PINK1/Parkin-dependent mitophagy [20]. However, the effect of LIPUS on the mitochondria of diabetic ECs is still unknown.

Here, we used an in vitro model of diabetes induced by different doses of AGES and observed changes in human CCECs following LIPUS treatment, including effects on tube formation and apoptosis. RNA-seq and expression studies were performed to explore its deep mechanism. In addition, 3-methyladenine (3-MA) was used to verify the role of mitophagy promoted by LIPUS in improving endothelial injury caused by AGES. These findings provide an experimental basis for guiding clinical practice.

## 2. Results

### 2.1. Mitochondrial Dysfunction and Apoptosis Are Observed in the Corpus Cavernosal Endothelium of Dmed Patients

ED is induced by the detrimental actions of high glucose levels on endothelial cells [3]. To obtain an understanding of endothelial dysfunction in diabetes, we examined the corpus cavernosum endothelial cells (CCEC) of normal and DMED patients using scanning electron microscopy (SEM). In normal males, the ECs were neatly arranged with narrow gaps in between cells, resembling a tiled pattern. In contrast, the CC endothelium of DMED patients was recognizably swollen and disorganized, forming a papillary bulge on the surface. Parts of the endothelium had been lost, resulting in exposure of subcutaneous collagen fibers to the vascular lumen (Figure 1A). We also used transmission electron microscopy (TEM) to examine subcellular structural changes in DMED ECs and found that the mitochondria showed obvious morphological changes, including increased size and decreased density (Figure 1B).

In addition, we used Von Willebrand Factor (VWF) to label ECs in order to investigate the apoptosis ratio between different cell types in DMED patients. VWF-positive DMED CCECs showed a significant increase in the percentage of γH2AX variant histone (γH2AX), a DNA damage double-stranded marker which could be detected during meiosis in germ cells or during apoptosis in non-germ cells, compared with those of normal CC (17.61% in DMED CC vs. 52.00% in normal CC) (Figure 1C,D). In addition, although VWF-negative cells also showed increased levels of apoptosis in the DMED CC, the overall proportion of apoptotic VWF-negative cells was far lower than that of apoptotic EC cells (6.33% in VWF-negative cells and 16.00% in VWF-positive cells) (Figure 1C,E). These results indicated that although the diabetic state damaged all CC cells, ECs were more susceptible than other cells.

### 2.2. Ages Treatment Induces Apoptosis and Dysfunction of Human Ccecs In Vitro

Hyperglycemia caused by diabetes mainly exerts harmful effects by inducing the production of AGES. To further explore the effect of AGES on CCECs, we cultured CCECs with low (20 µg/mL, LAGES group) and high (100 µg/mL, HAGES group) concentrations of AGES. Examination by light microscopy revealed that AGES had dose-dependent effects on the growth rate, cell morphology, and colony characteristics of CCECs. After AGES treatment, cell morphology changed from oval to fibroblast-like, while the growth pattern of cell colonies altered from cobblestone-like to scattered (Figure 2A); proliferation also decreased with high AGES concentration (Figure 2B). Moreover, we found that AGES treatment induced apoptosis in CCECs: the addition of AGES at 20 µg/mL increased the early apoptosis ratio from 1.4% to 6.7% (*p* < 0.001) and increased the late apoptosis ratio from 5.4% to 19.5% (*p* < 0.001), while the increased AGES concentration (100 µg/mL) led to further increases in the level of apoptosis (Figure 2C–E).

Examination of DMED CC tissue revealed localized endothelial shedding (Figure 1A); repair of such injuries depends on the proliferation and migration of adjacent ECs. We found that AGES treatment also inhibited the migratory ability of CCECs through the cell scratch test (63.3% in NC vs. 54.5% in LAGES group vs. 43.0% in HAGES group, *p* < 0.0001) (Figure 2F,G). Angiogenesis is another important function of CCECs required for the repair of damage to CC tissues. To test the effects of AGES on the angiogenic ability of CCECs, we performed assays to analyze both tube formation and vascular bud length. Firstly, analysis of tube formation by CCECs showed that total mesh area (positively correlated with tube formation) decreased from 175,209 in NC to 93,784 in LAGES group to 46,784 in HAGES group (*p* < 0.0001; Figure 2H,I). The second assay examined the spontaneous formation of vascular buds and the lengths of which represent the angiogenic ability of CCECs. We found a decrease in the ratio of vascular bud length to the radius of cell mass in AGES treatment groups (1.4 in NC vs. 1.2 in LAGES group vs. 1.0 in HAGES group, *p* = 0.0005) (Figure 2J,K). Taken together, the above results indicated that AGES treatment not only disrupted CCEC function but also caused a loss of cell numbers.

### 2.3. AGES Treatment Perturbs the Transcriptome in Human CCECs

Although the EC dysfunction mediated by AGES has been studied [3], a comprehensive understanding at the molecular level is still lacking. Therefore, we explored the overall changes in the CCEC transcriptome induced by AGES treatment. Using set log_2_fold change > 2 and *p* < 0.05 as the threshold yielded 417 up-regulated genes and 528 down-regulated genes (Figure 3A). Consistent with the TEM analysis of DMED CCECs, the transcriptomic changes revealed a significant enrichment of mitochondrial dysfunction and oxidative phosphorylation pathways, indicating that EC mitochondria were impaired in morphology and function. Disruption of energy metabolism and excessive ROS caused by mitochondrial dysfunction are important reasons for cell apoptosis and senescence, which was confirmed by an enrichment of the sirtuin signaling and antioxidant action of vitamin C pathways. Other pathways, such as the airway pathology in chronic obstructive pulmonary disease and immunogenic cell death signaling pathways, indicated abnormal inflammatory reactions in the diabetic state. In addition, endothelin-1 signaling and relaxin signaling were significantly activated and inhibited, respectively, suggesting that the endothelial regulation of smooth muscle contraction and relaxation was abnormal (Figure 3B).

To further describe the changes in regulatory networks induced by AGES treatment, we used GENIE3 to predict altered networks of transcription factors and their target genes. A further comparison of differentially expressed genes revealed a significant up-regulation of some transcription factors at the central nodes of regulatory networks, such as ZNF580, TCF12, and RFX7, while others, such as HMGB2, MYC, and SMAD4, were significantly down-regulated (Figure 3C). ZNF580, TCF12, HMGB2, and MYC have been shown to be closely related to angiogenesis in many studies [21,22,23,24]. In addition, Ingenuity Pathway Analysis (IPA) identified an enrichment of the inhibition of angiogenesis by TSP1 pathway in the AGES treatment group (Figure 3B). Several angiogenesis-related genes showed significantly different expression patterns in the AGES-treated and untreated groups (NC) (Figure 3D).

### 2.4. LIPUS Treatment Rescues AGES-Induced Impairment of Angiogenesis and Transcriptional Changes in CCECs

In animal models and clinical studies, LIPUS has been shown to improve erectile function and promote structural repair in the CC [16,25]. As AGES treatment significantly disrupted angiogenic function in CCECs, we assayed tube formation and vascular bud growth in CCECs treated with AGES and/or LIPUS to evaluate whether LIPUS stimulation could counteract the effects of AGES treatment. The results showed that LIPUS stimulation promoted CCEC angiogenesis in a dose-dependent manner, with an optimal therapeutic intensity of 100 mW/cm^2^ (Figure 4A–D).

It is worth noting that LIPUS activates downstream signals through mechanical force conduction, while excessive energy may cause cell damage. We stimulated CCECs using LIPUS treatment at 100 mW/cm^2^ and performed transcriptomic sequencing to evaluate the molecular effects in each pathway. LIPUS treatment yielded 583 up-regulated genes and 469 down-regulated genes (Figure 5A). Further comparison among the three groups (untreated, AGES-treated, and AGES+LIPUS-treated) showed that the transcriptomic status of the AGES+LIPUS-treated group was similar to that of the untreated group (Figure 5B). Furthermore, the selection of the top 2000 genes (by expression level) to perform principal component analysis (PCA) and generate 3D plots confirmed that LIPUS treatment promoted a more normal transcriptional state in AGES-treated CCECs (Figure 5C).

When performing the gene ennrichment analysis base on the differentially expressed genes (DEGs) between AGES-treated and AGES+LIPUS-treated groups, we found pathways related to cellular states such as “Cell aging”, “Cell cycle”, and “Apoptosis”, and pathways related to mitochondria such as “mitochondrion organization”, “apoptotic mitochondrial changes”, and “Mitophagy-animal” were significantly enriched (Figure 5D,E). This result indicated that the LIPUS treatment changed cellular and mitochondrial states via mitophagy or other mechanisms.

### 2.5. LIPUS Restores Mitochondrial Morphology and Reduces Mitochondrial ROS Production Induced by AGES in CCECs

Recent studies have demonstrated that mitophagy eliminates damaged and dysfunctional mitochondria, stabilizes mitochondrial structure and function, reduces mitochondrial ROS production, and maintains cell survival and growth [26,27]. To verify the pathway enrichment results of RNA sequencing, we used MitoSOX™ Red to label mitochondrial ROS and found that the AGES treatment increased the levels of mitochondrial ROS in a dose-dependent manner, while LIPUS stimulation rescued the effect of AGES treatment (Figure 6A,B).

Mitochondrial dynamics play an important role in diabetes establishment and progression as well as in associated conditions [28]. We observed that treatment with a high level of AGES (100 µg/mL) induced significant changes in the morphology of CCEC mitochondria, altering their state from fusion to fission, as evaluated by form factor (indicated the fusion level) (3.0 in untreated group, vs. 1.2 and AGES-treated group, *p* < 0.0001). However, LIPUS treatment partially reversed this change: in cells treated with 100 µg/mL AGES, stimulation by 100 or 200 mW/cm^2^ LIPUS significantly increased the mitochondrial form factor from 1.2 in the AGES-treated group to 2.5 in the AGES+LIPUS-treated group (*p* = 0.0025) or 2.3 (*p* = 0.012), respectively (Figure 6C,D).

### 2.6. LIPUS Activates Mitophagy in CCECs In Vitro

To validate the level of mitophagy, we performed ICC co-staining using Mito-Tracker^®^ Red and anti-LC3 antibody. LC3 is an autophagy marker involved in the formation of autophagosomes. Although AGES treatment increased the intensity of LC3 staining in the cytoplasm, staining was unchanged in mitochondrial regions. However, LIPUS treatment significantly increased the intensity of LC3 staining in both whole-cell cytoplasmic and mitochondrial regions, indicating activation of both autophagy and mitophagy (Figure 7A,B). Furthermore, we examined the process of mitophagy by TEM. Under normal culture conditions, the structure of mitochondria in CCECs was intact and their internal morphology was clearly visible. However, while stimulation with AGES induced morphological changes, high levels of lysosomal-mitochondrial fusion were only observed in the LIPUS-stimulated group (Figure 7C).

### 2.7. Inhibition of Mitophagy Attenuates the Therapeutic Effects of LIPUS

Although we found that LIPUS promoted mitophagy in AGES-treated CCECs, the mechanism of action of LIPUS may be multi-faceted [29,30]. Therefore, we used 3-MA, a macroautophagy/mitophagy inhibitor, to verify the role of mitophagy in LIPUS-mediated protection against AGES-induced endothelial dysfunction. First, we used WB to verify the inhibitory efficiency of 3-MA, and the results showed that LIPUS significantly up-regulated the expression levels of autophagy (LC3 and p62) and mitophagy (PARKIN) proteins in mitochondria, while 3-MA treatment could inhibit this effect (Figure 8A). We then used common endothelial cell markers (CD31) and CCEC-specific marker (KIT) to explore the effect of LIPUS-promoted mitophagy on the maintenance of endothelial cell status, and found that LIPUS restored the decreased CD31 expression induced by AGES, while 3-MA treatment also inhibited the effect of LIPUS (Figure 8B).

When focused on cellular state and function CCECs, LIPUS reduced AGES-induced apoptosis in CCECs; this therapeutic effect was significantly inhibited by 3-MA treatment (Figure 8C–E). Tube formation ability was similarly affected: combining LIPUS with 3-MA treatment resulted in a lower total mesh area than LIPUS treatment alone, and the results showed no significant difference when compared with the AGES-treated group (Figure 8F,G).

## 3. Discussion

The incidence of diabetes and its complications is increasing year by year, and it is estimated that by 2025, there will be 300 million patients worldwide [31]. Diabetics are three times more likely to develop ED, a common complication of diabetes [32]. Although the pathogenesis of diabetic ED is multifactorial, endothelial dysfunction is considered to be an important pathological basis for this condition [33]. Insufficient production of nitric oxide in the ECs of diabetic patients inhibits the mechanism of cGMP-mediated smooth muscle relaxation and causes insufficient blood supply to the CC. Furthermore, in the diabetic state, apoptosis of ECs is increased, while the expression of cell connexins such as GJA1, CLDN5, and ZO-1 is decreased, leading to vascular leakage and failure to maintain erections [34]. In addition to impairing endothelial function, long-term diabetes can also cause extensive organic changes in the CC tissue, mainly manifested by morphological changes, decreased numbers of ECs, and increased levels of collagen [9].

Chronically elevated glucose levels have been reported to induce the formation of AGES, which are known to interfere with endothelial nitric oxide production by directly inhibiting the phosphorylation of endothelial nitric oxide synthase (eNOS) [35]. Previous studies have generally focused on the effect of AGES or high glucose on the endothelium-smooth-muscle-erectile axis. However, EC function and survival also affect the stability of the CC structure, which may be the reason why many diabetic ED patients do not respond well to phosphodiesterase type 5 inhibitors (PDE5I) [36]. Bioinformatic analysis based on proteomics studies carried out on human umbilical vascular endothelial cells (HUVECs) revealed that the differentially regulated proteins were involved in various processes such as apoptosis and oxidative stress [37]. However, as our previous studies have shown that CCECs are more susceptible to diabetic damage than the vascular endothelium [9], studies based on HUVECs may not accurately reflect the diabetic conditions of CCECs and the cause of DMED. Therefore, we comprehensively evaluated the effects of AGES treatment on CCECs using RNA-seq and identified several highly enriched terms related to mitochondria. In addition, electron microscopy of CC tissue from patients as well as analysis of cultured CCECs showed apoptosis, shedding, and mitochondrial structural disorders of CCECs to be important features of DMED. Indeed, mitochondrial damage, increased ROS levels, and endothelial dysfunction induced by AGES stimulation are likely to be continuous changes caused by the same pathological process [38]. Therefore, when considering treatment for DMED, in addition to controlling blood glucose and AGES at the source, protecting and repairing the mitochondrial function of CCECs may have a more direct therapeutic effect.

Mitophagy is an autophagic response that specifically targets damaged, and hence, potentially cytotoxic, mitochondria. In recent years, mitophagy has been proven to play a protective role against the development of insulin resistance and diabetic conditions in cardiovascular, muscle, adipose, and kidney tissue [13,14,39]. Current studies indicate that LIPUS, which transmits mechanical energy, can safely and effectively treat ED, resulting from a variety of causes, and can reverse the pathological changes of the CC [16,40,41]. LIPUS also potently activates PINK1/Parkin-dependent mitophagy and plays a role in promoting β-cell resistance to diabetes-related hyperlipotoxicity [41]. Our study found that LIPUS promoted mitophagy and reduced mitochondrial ROS production as well as mitochondrial fission in a dose-dependent manner. In addition, LIPUS promoted CCEC angiogenesis, which is essential for the equilibrium between injury and endogenous regeneration capacity; disturbance of this balance in favor of the former results in the irreparable loss of the functional integrity of the endothelium and the exacerbation of vascular disease [42]. In addition, the mitophagy inhibitor 3-MA attenuated the protective effect of LIPUS in CCECs pre-treated with AGES, demonstrating that the effect of LIPUS was mitophagy-dependent. However, as a mechanism of cellular self-protection, the specific process by which LIPUS stimulates mitophagy requires further investigation.

However, there are some limitations to this study. For example, all CC tissue was derived from penile cancer patients. Although these patients had normal erectile function and tumor tissue was avoided during sampling, tumor compression or paracrine effects may still affect adjacent normal tissues, especially the morphology and function of CCEC. So, future tissue from healthy donors remains essential to confirm the conclusions of this study. Another limitation is that because DMED CC tissue is very precious and difficult to obtain, there were only two repeats in the Figure 1 part of our study. Although the conclusions of this part were verified by subsequent in vitro experiments and functional experiments, it is also necessary to further expand the sample size for verification.

In summary, we demonstrated that LIPUS restored AGES-induced CCEC damage by reducing apoptosis and promoting angiogenesis via protecting mitochondrial structure and function. LIPUS activated mitophagy by upregulating mitochondrial LC3, p62, and PARKIN, reduced levels of mitochondrial ROS, and blocked the AGES-induced changes in the mitochondrial state from fusion to fission. Therefore, our study provides theoretical support for LIPUS-based therapy and paves the way for further in vivo studies to explore improved treatment options for DMED.

## 4. Materials and Methods

### 4.1. Contact Information for Reagent and Resource Sharing

Requests for further information, resources, and reagents should be directed to and will be fulfilled by the Lead Contact or the first author, Yuzhuo Chen (chenyzh223@mail.sysu.edu.cn).

### 4.2. Experimental Models and Subject Details

The experiments performed in this study were approved by the Ethics Committee of The Fifth Hospital Affiliated Sun Yat-sen University (License No. 2022-L170-1). CC samples were obtained from the tumor margin of penile carcinoma resection, all patients reported good stimulated erections and early morning erections, under the age of 50. LIPUS Device (WBL-ED, ED-ZLT-F treatment head) was from Wanbeili Medical Device Co., LTD., Beijing, China. In the CCECs in vitro experiments, CCECs were treated with low (20 µg/mL, LAGES group) or high (100 µg/mL, HAGES group) concentrations of AGES with or without 10 mM 3-MA for 48 h [43,44,45], then, were stimulated with LIPUS three times, 5 min each time with a 1 min interval, according to previous studies [20,25].

### 4.3. Isolation and Culture of CCECs

Fresh CC tissue samples of approximately 1 × 1 cm size were first immersed in 15 mL PBS with vigorous shaking to remove residual blood cells. Then, tissues were cut into pieces of 1 × 1 mm size and enzymatically digested with 2.5 mg/mL collagenase type I (17104-019, gibco, US), 4 mg/mL collagenase type IV (17018029, gibco), 0.1 mg/mL neutral protease, and 2 mg/mL DNase I (#AMPD1, Sigma-Aldrich, US) at 37 °C for 20 min. Subsequently, the cell suspension was filtered through a 40 mm nylon mesh, and the cells were sorted by MACS with a Dead Cell Removal Kit (130-090-101, Miltenyi Biotec, Germany) to remove dead cells. Briefly, 10^6^ cells were co-incubated with 20 µL MicroBeads at 4 °C for 30 min. After blocking and washing, magnetic separation with LS Columns (130-042-401; Miltenyi) and MidiMACS Separator (130-042-302; Miltenyi) was performed, retaining cells that were not adsorbed by the column. CCECs were cultured with EGM-2 medium and cell cloning began at approximately day 5 of culture, then fibroblast inhibitors were added for 7–10 days. When isolated colonies began to fuse with each other, the EC colonies were marked and digested using a clonal cylinder (C7983-50EA; Sigma-Aldrich), and the EC suspension could be further purified using CD31 magnetic beads (according to their purity) (130-091-935; Miltenyi).

### 4.4. Immunocytochemistry (ICC) and Immunohistochemical (IHC) Staining

The CCECs with the ability of adherence growth were cultured in the cell slide. After treatment, cells were washed with phosphate-buffered saline (PBS) and fixed with 4% paraformaldehyde (PFA). The cells were blocked with 5% bull serum albumin (BSA) for 1 h and incubated with relevant primary antibodies overnight at 4 ℃. These cells were further incubated with secondary antibodies for 2 h. Cell nuclei were labelled by Hoechst33342.

Mitochondrial ROS staining of CCEC was performed using 0.5 µg/mL MitoTracker™ Red CMXRos (M36007, Invitrogen, US). The cytoskeleton staining of CCEC was performed using 1:500 diluted FITC-Phalloidin (G1028, Servicebio, China). Mitochondrial staining of CCEC was performed using 0.1 µg/mL MitoTracker™ Red CMXRos (M7512, Invitrogen, US) by incubating 30 min, and mitochondrial morphology (fusion and fission) scoring was performed using Mitochondria Analyzer function of ImageJ software.

Sections were performed with rehydration, antigen retrieval, and blocking, and then incubated with appropriate primary antibodies at 4 °C overnight. Sections were further incubated with a secondary antibody for 2 h at room temperature. Nuclei were labelled with Hoechst33342 by incubating tissue sections for 15 min [46]. Images were captured with an OLYMPUS IX83 confocal microscope. Fluorescence intensity measurements were processed with Adobe Photoshop CC 2018 software v 23.0.2.

Primary antibodies were used for ICC and IHC in this study, including Anti-VWF Rabbit pAb (GB11020, Servicebio), Anti-phospho-Histone H2AX (Ser139) Antibody (05-636-I, Sigma), and Anti-VWF Rabbit pAb (16931, Life). Alexa Fluor 488 donkey anti-rabbit IgG (A21206, Thermo Fisher, US), Alexa Fluor 555 donkey anti-rabbit IgG (A31572, Thermo Fisher), Alexa Fluor 488 donkey anti-mouse IgG (A21202, Thermo Fisher), and Alexa Fluor 555 donkey anti-mouse IgG (A31570, Thermo Fisher) were used as secondary antibodies.4.5. Protein extraction and Western blotting were carried out.

Briefly, CCECs that treated AGES, LIPUS and/or 3-MA were collected from 6-well plates and lysed with RIPA lysis buffer (Dingguo, WB-0071) on ice. Protein extracts were heated at 95 °C with 5× SDS PAGE Loading Buffer (Dingguo, WB-0091) for 10 min. A total of 30 mg of protein was electrophoresed on SDS-PAGE and transferred to polyvinylidene difluoride membranes according to the procedure previously described [47]. After blocking with 5% skim milk, membranes incubated the first antibodies including Anti-LC3A/B Rabbit pAb (GB11124, Servicebio), Anti-SQSTM1/p62 Rabbit pAb (GB11239-1, Servicebio), Anti-Parkin Rabbit pAb (GB113802, Servicebio), Anti-COX IV Rabbit pAb (GB11250, Servicebio), Anti-CD31 Rabbit pAb (GB11063-2, Servicebio), Anti-c-kit Rabbit pAb (GB113799, Servicebio), and Anti-beta Actin antibody (GB15003, Servicebio) at 4 °C overnight and second antibodies at room temperature for 1 h. After being washed three times with TBS-T, the result of Western blotting was detected with Gel Imaging System (Chemi-Doc XRS, BIO-RAD, USA) and analysis with Adobe Photoshop CC 2018 software v 23.0.2.

### 4.5. Endothelial Cell Functional Experiments

Cell migration: CCECs were seeded in a 6-well plate at a concentration of 1 × 10^6^ per well. After two days, all cells completed contact with each other, and AGES and/or LIPUS treatment were performed according to the group. After 24 h, a 10 µL pipetting gun tip was used to make scratches and then the shed cells were removed. The results were observed 12 h later. The area ratio of cell migration regions and whole scratch regions in each group were recorded and counted.

Tube formation: CCEC cells were re-suspended in each group of medium supplemented (Lonza, CC-3202, Switzerland) at a concentration of 200 μL, and then seeded on µ-Slide (81506, ibidi, Germany) with pre-added 10 μL of Matrigel (256230, Corning, US). After 2 h, CCECs were fixed with PFA, and then incubated with FITC-phalloidin (G1028-50UL, Servicebio, China) for 30 min. Images were captured with an OLYMPUS IX83 confocal microscope. The tube formation analysis, including the “total meshes area score”, was performed with Angiogenesis Analyze function of ImageJ software v 1.8.0.172 [48].

Vascular bud length: CCECs were seeded in AggreWell (34411, STEMCELL, Canada) at a concentration of 3 × 10^5^ per well; after 12 h, the cell mass was re-suspended and seeded on µ-Slide (81506, ibidi) with the pre-added 10 μL of Matrigel (256230, Corning, US) for 6 h. Next, cell supernatant was replaced with conditioned medium (Lonza, CC-3202, Switzerland) for 24 h and fixed. The ratio of cell mass diameter to the distance from the center of the cell mass to the farthest end of the vascular bud in each group was recorded and counted.

### 4.6. Cell Apoptosis Analysis

Annexin V-FITC apoptosis detection kit (A211-01, Vazyme Biotech, China) was used to measure cell apoptosis according to the manufacturer’s protocol. In brief, CCECs were collected by EDTA-free trypsin, washed with ice-cold phosphate-buffered saline twice, and resuspended in 100 μL of binding buffer. Next, the cells were incubated with 5 μL of Annexin V-FITC and 5 μL of PI staining solution in the dark for 10 min, and then diluted with binding buffer to 500 μL. Cell apoptosis was subsequently analyzed by flow cytometry (Beckman Coulter, Inc., Brea, CA, USA); the fluorescence signals of Annexin V and PI conjugate were detected in fluorescence intensity channels B525-FITC and Y585-PE without a compensation set, respectively. The acquisition speed was 60 μL/min. The apoptotic rates were measured by FlowJo software, version 10.6.1 (FlowJo LCC, Becton Dickinson, Ashland, OR, USA).

### 4.7. RNA-Seq of CCECs In Vitro

The 1 × 10^5^ CCECs cells were seeded in a six-well microplate. After 24 h, AGES at a final concentration of 100 µg/mL was added or not, and a standard dose of LIPUS stimulation was performed or not. The medium was changed every other day during culturing. After 2 days, all cells were harvested with TRIzol (15596026; Thermo Fisher) for mRNA sequencing. Library construction and sequencing were performed by Sinotech Genomics Co., Ltd. (Shanghai, China). Total RNA was isolated using RNeasy mini kit (74904; Qiagen). The poly(A)-containing mRNA molecules were purified using poly(T)-oligo-attached magnetic beads. Then, the mRNA underwent fragmentation, reverse transcription, purification, and enrichment. Clusters were generated by cBot with the library diluted to 10 pM, and then were sequenced on the Illumina NovaSeq 6000 (Illumina, USA). Paired-end sequence files (fastq) were mapped to the reference genome (GRh38-1.2.0) using HISAT2. Differential expression analysis for mRNA was performed using the R package edgeR. Differentially expressed RNAs with |log_2_(FC)| value > 1 and q value < 0.05, considered significantly modulated, were retained for the volcano plot and GO analysis.

### 4.8. Statistics and Reproducibility

Data in Figure 2D,F,H,J; Figure 4B,D; Figure 6D,E,F; and Figure 7B,C,E were shown as mean ± SD. Data were first checked for normality with the D’Agostino-Pearson and Kolmogorov–Smirnov test, and statistical significance was calculated with the Kruskal–Wallis analysis and Dunn’s test (for the result in Figure 2D,F,H,J, Figure 4B,D, Figure 6D,E,F and Figure 7B,C,E unpaired, two-tailed), and the chi-square test (Figure 1D,E) by GraphPad Prism 9.0.0 software. The confidence interval was 95%. Results were considered significant at *p* < 0.05. Statistical parameters are reported in the respective figures and figure legends. IHC and ICC staining for each disease type in this study were based on at least two biological repeats (five penile tumor patients with normal erectile function and two DMED patients). Scanning electron microscope and transmission electron microscope image (Figure 1A,B) were based on two biological repeats.

## Figures and Tables

**Figure 1 ijms-23-14887-f001:**
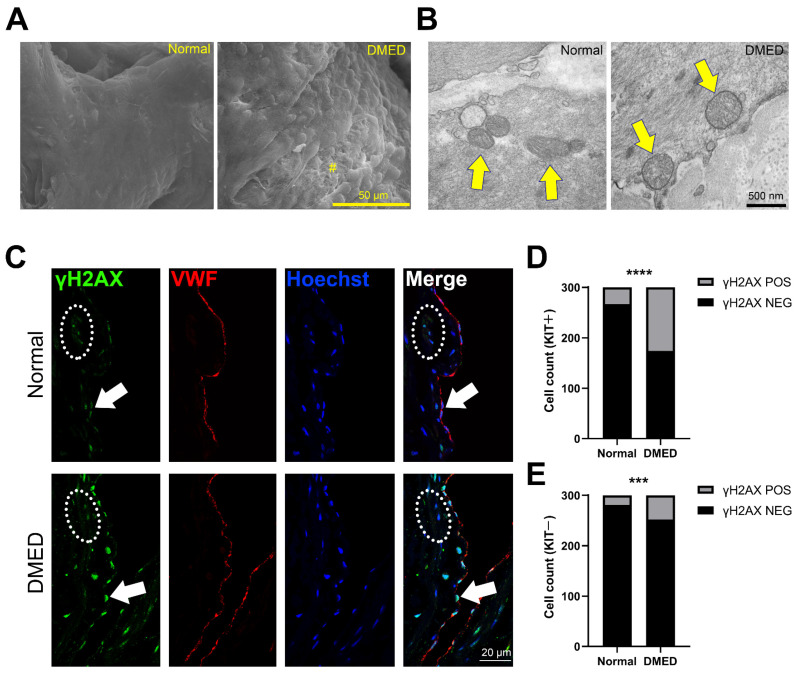
The pathological change of CCECs in DMED patients. (**A**) Scanning electron microscope image of normal and DMED cavernosal trabecular endothelium. The scale bar represents 50 µm. “#” labeled the region where the endothelium is detached. (**B**) Transmission electron microscope image of normal and DMED CCECs. Mitochondria are marked by yellow arrows. The scale bar represents 500 nm. (**C**) Immunofluorescence co-staining of VWF (red) and γH2AX (green) in CC paraffin sections. The white arrows point to γH2AX and VWF double-positive CCECs, and VWF-negative non-endothelial cells are marked in the dotted box. The scale bar represents 20 µm. (**D**,**E**) The statistical analysis of the γH2AX positive and negative ratio in CCECs (**D**) and other CC somatic cells (**E**). ***, *p* < 0.001; ****, *p* < 0.0001. n = 300 cells from 5 normal CC samples and 300 cells 2 DMED CC samples. DMED: Diabetes mellitus-induced erectile dysfunction; CCEC: corpus cavernosum endothelial cells.

**Figure 2 ijms-23-14887-f002:**
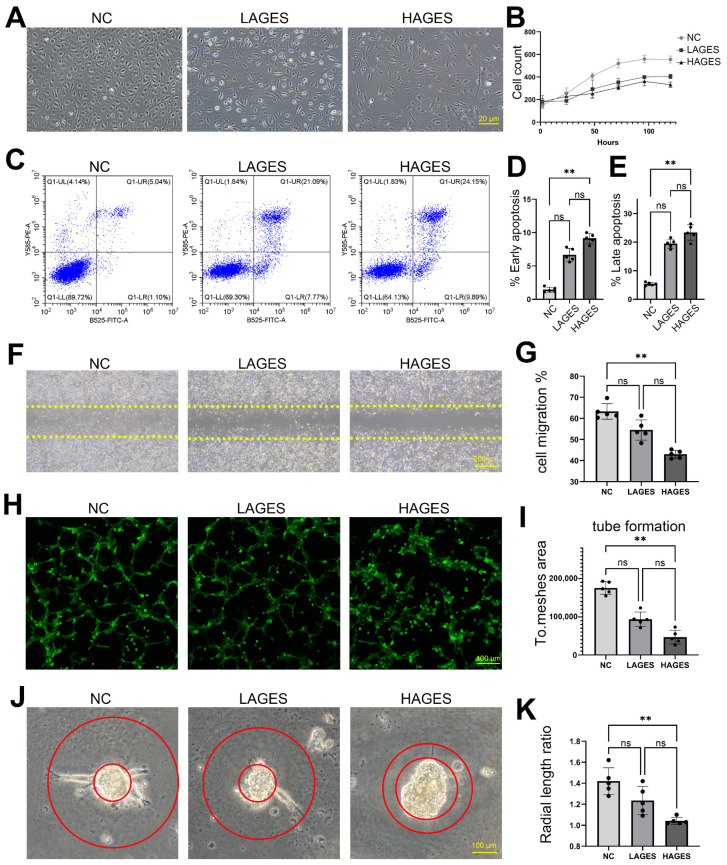
The effect of AGES treatment on CCECs. (**A**) Morphological changes of CCECs treated with different concentrations of AGES under light microscope in vitro. The scale bar represents 20 µm. (**B**) The cell count per 40× magnification region of CCECs treated with different concentrations of AGES for 5 days. (**C**–**E**) Flow cytometry demonstrates the different concentrations of AGES treatment on the early apoptosis (**D**) and late apoptosis (**D**) of CCECs. The existence of Annexin V (*X*-axis) and the nuclear staining of PI (*Y*-axis) by flow cytometry are shown. n = 5 independent repeats. ns, not significant; **, *p* < 0.01. (**F**,**G**) CCECs migration change after different concentrations of AGES treatment. n = 5 independent repeats. Data are shown as mean ± SD. ns, not significant; **, *p* < 0.01. (**H**,**I**) Co-staining of phalloidin (green) and Hoechst33342 (blue), showing typical changes of CCECs tube formation after different concentrations of AGES treatment. The total mesh area score is used to represent the level of tube formation in each group. n = 5 independent repeats. Data are shown as mean ± SD. ns, not significant; **, *p* < 0.01. (**J**,**K**) The radial length of CCEC mass vascular bud change after different concentrations of AGES treatment. n = 5 independent repeats. Data are shown as mean ± SD. ns, not significant; **, *p* < 0.01. NC: negative control (0 µg/mL AGES); LAGES: low level (20 µg/mL) of advanced glycation end products-treated group; HAGES: high level (100 µg/mL) of advanced glycation end products-treated group.

**Figure 3 ijms-23-14887-f003:**
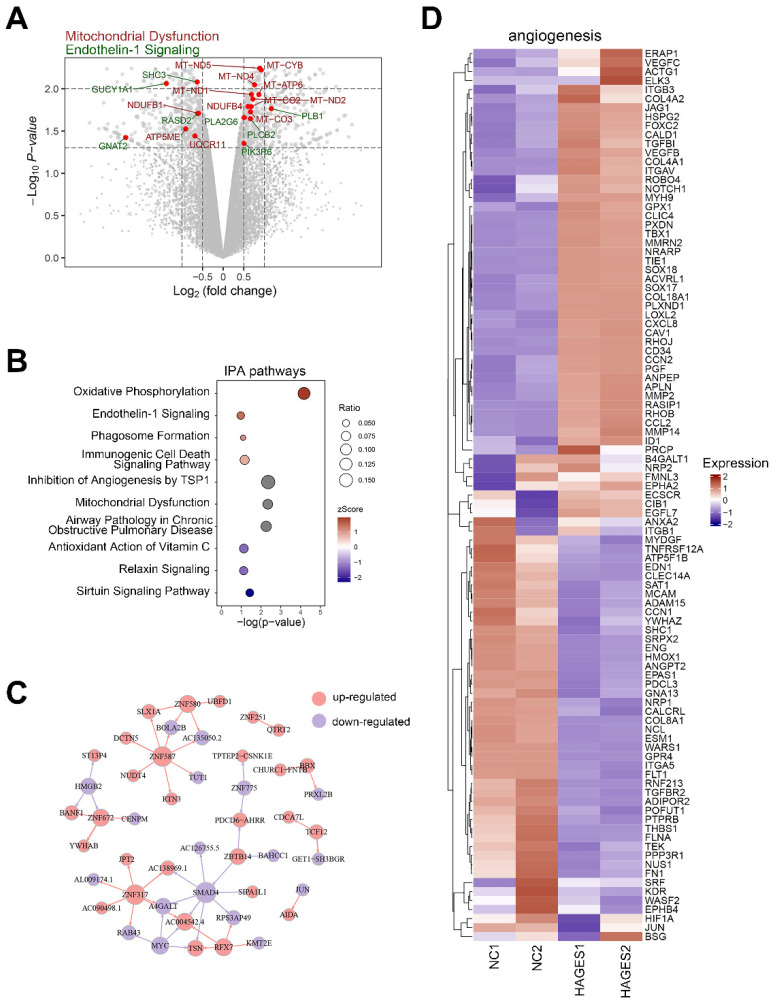
The transcriptome changes of CCECs after AGES treatment. (**A**) Volcano plot showing the DEGs between CCECs treated with AGES and negative controls by RNA sequencing. Genes belonging to the GO terms “Endothelin-1 Signaling” and “Mitochondrial dysfunction” are labelled with green and brown, respectively. (**B**) Bubble diagram of the enriched IPA pathway of CCECs after being treated with AGES. A gradient of light blue to red indicates inhibition to activation of the term. The size of the bubble indicates the *p*-value from high to low. (**C**) The transcription factors and target genes network of CCECs after being treated with AGES. The direction of the arrow indicates that the transcription factors point to the target genes, and the color of the node indicates up-regulation (red) or down-regulation (blue). (**D**) Heatmap shows the expression level of genes belonging to the angiogenesis GO term between CCECs treated with AGES and negative controls. A gradient of light blue to red indicates the down-regulation to up-regulation of the genes. NC: negative control (0 µg/mL AGES); HAGES: high level (100 µg/mL) of the advanced glycation end products-treated group.

**Figure 4 ijms-23-14887-f004:**
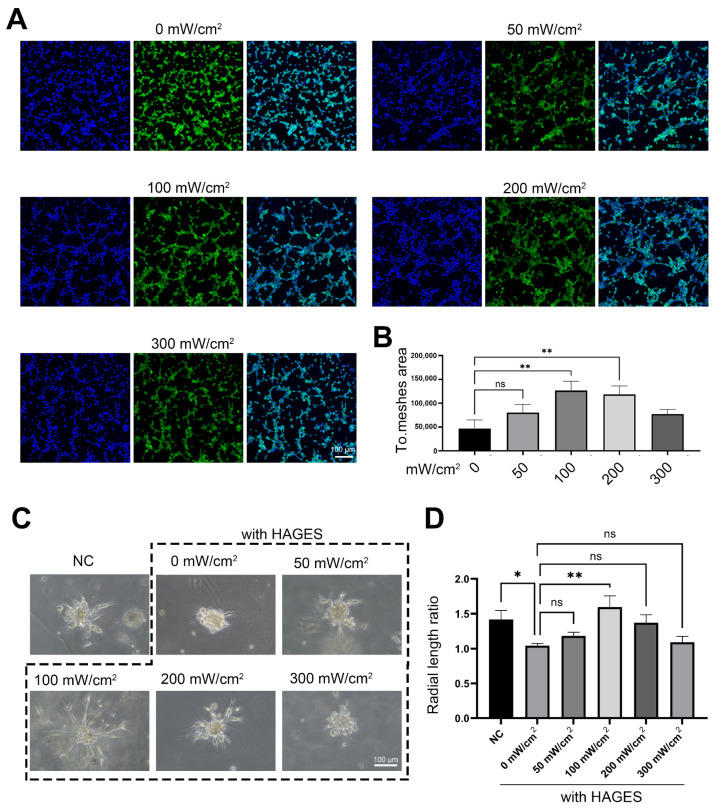
The effect of LIPUS treatment on CCECs under the AGES state. (**A**,**B**) Co-staining of phalloidin (green) and Hoechst33342 (blue), showing typical changes of CCECs tube formation after different doses of LIPUS stimulation under AGES treatment state. The total mesh area score is used to represent the level of tube formation in each group. n = 5 independent repeats. Data are shown as mean ± SD. ns, not significant; **, *p* < 0.01. (**C**,**D**) The radial length of CCEC mass vascular bud change after different doses of LIPUS stimulation with or without AGES treatment. n = 5 independent repeats. Data are shown as mean ± SD. ns, not significant; *, *p* < 0.05; **, *p* < 0.01. NC: negative control (0 µg/mL AGES); HAGES: high level of advanced glycation end products (100 µg/mL).

**Figure 5 ijms-23-14887-f005:**
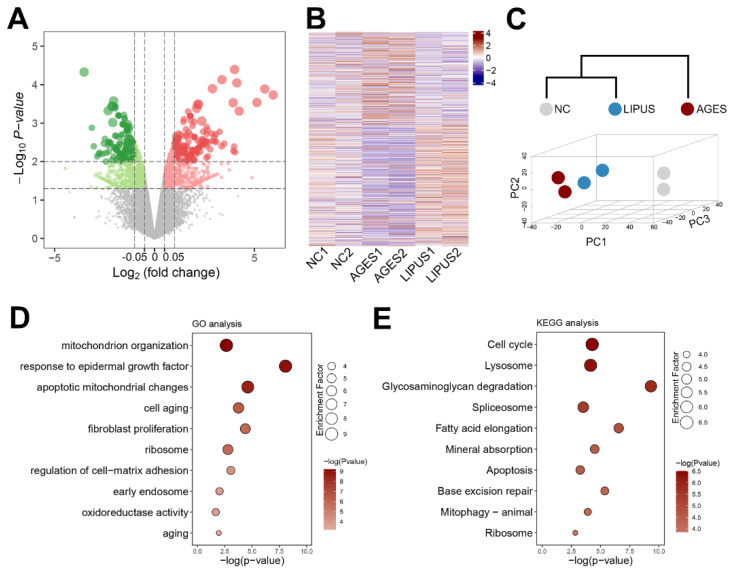
The transcriptome changes of CCECs after LIPUS treatment. (**A**) Volcano plot showing the DEGs between CCECs treated with AGES and negative controls by RNA sequencing. A gradient of green to red indicates the down-regulation to up-regulation of the genes. (**B**) Heatmap shows the expression level of DEGs among NC, AGES treatment, and AGES with LIPUS treatment groups. A gradient of blue to red indicates the down-regulation to up-regulation of the genes. (**C**) PCA 3D mapping plot and cluster analysis dendrogram of CCEC samples in each group. (**D**,**E**) Bubble diagram of the enriched GO (**D**) or KEGG (**E**) of CCECs. A gradient of light gray to red indicates inhibition to activation of the term. The size of the bubble indicates the *p*-value from high to low. NC: negative control (0 µg/mL AGES and 0 mW/cm^2^ LIPUS); AGES: advanced glycation end products (100 µg/mL)-treated group; LIPUS: low-intensity pulsed ultrasound and 100 µg/mL AGES-treated group.

**Figure 6 ijms-23-14887-f006:**
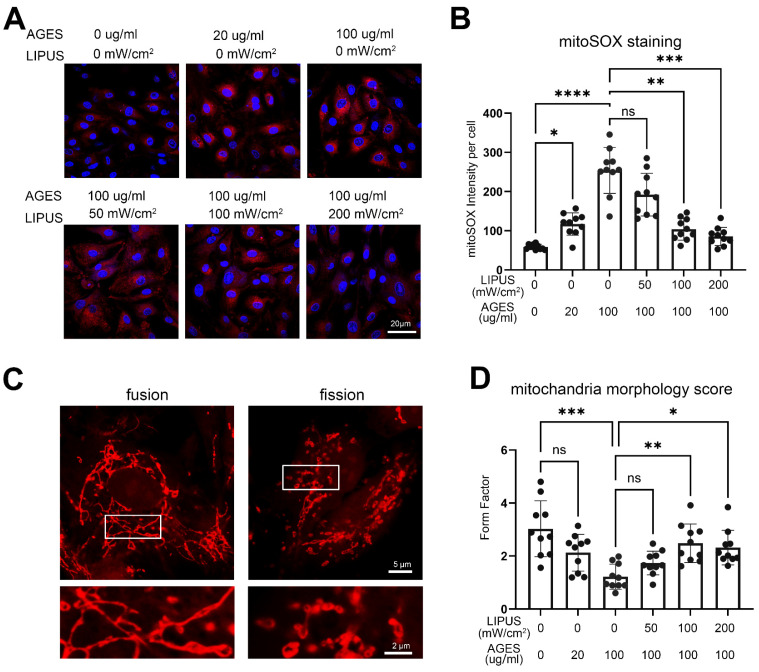
The effect of LIPUS treatment on CCECs mitochondria. (**A**) Immunofluorescence staining of mitoSOX (red) in CCECs after different doses of LIPUS stimulation under the AGES treatment state. The scale bar represents 20 µm. (**B**) The statistical analysis of the mitoSOX fluorescence intensity per CCEC. n = 10 regions from 5 independent samples. Data are shown as mean ± SD. ns, not significant; *, *p* < 0.05; **, *p* < 0.01; ***, *p* < 0.001; ****, *p* < 0.0001. (**C**) Immunofluorescence staining of mitoSOX shows the two typical morphologies of CCECs before and after LIPUS treatment. The lower panels are a magnification of the upper rectangle. The scale bar represents 5 µm (lower panel) and 2 µm (lower panel). (**D**) The statistical analysis of the Form Factor in each CCECs group. n = 10 regions from 5 independent samples. Data are shown as mean ± SD. ns, not significant; *, *p* < 0.05; **, *p* < 0.01; ***, *p* < 0.001. AGES: advanced glycation end products; LIPUS: low-intensity pulsed ultrasound.

**Figure 7 ijms-23-14887-f007:**
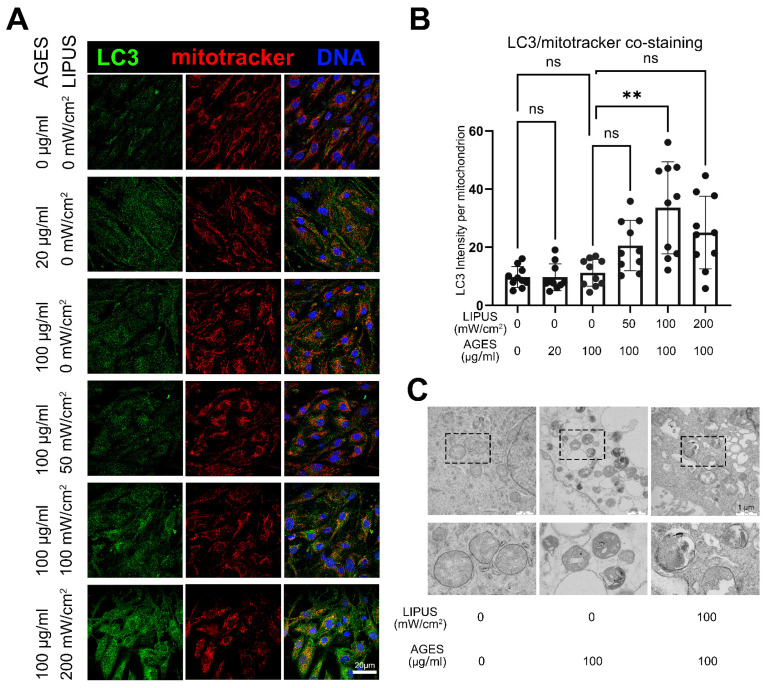
LIPUS treatment activated mitophagy in CCECs. (**A**) Immunofluorescence co-staining of mitotracker (red) and LC3 (green) in CCECs after different doses of LIPUS stimulation under the AGES treatment state. The scale bar represents 20 µm. (**B**) The statistical analysis of the LC3 fluorescence intensity in mitochondrial region. n = 10 regions from 5 independent repeats. Data are shown as mean ± SD. ns, not significant; **, *p* < 0.01. (**C**) Transmission electron microscope image of CCECs with AGES and LIPUS treatment. The scale bar represents 1 μm. AGES: advanced glycation end products; LIPUS: low-intensity pulsed ultrasound.

**Figure 8 ijms-23-14887-f008:**
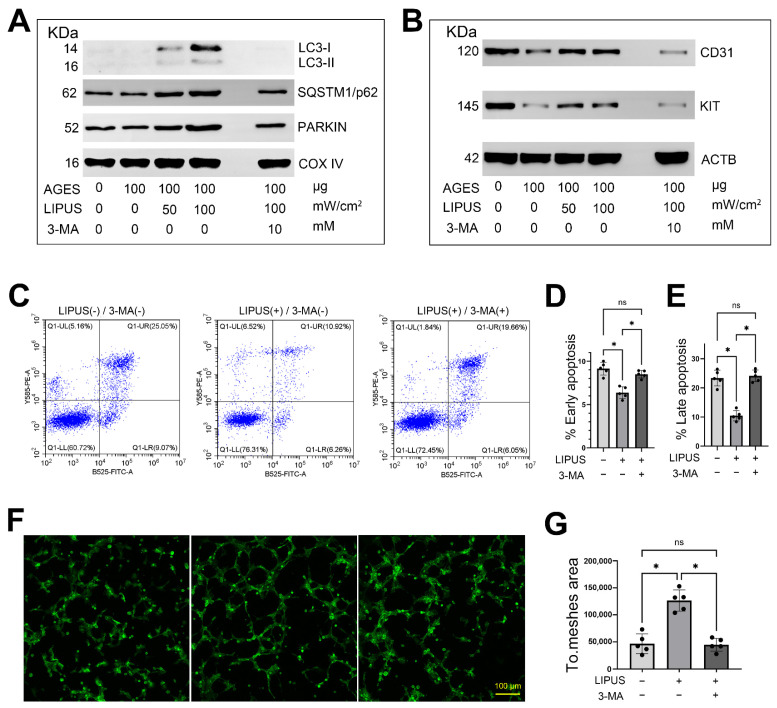
Inhibition of mitophagy attenuates the therapeutic effect of LIPUS. (**A**) Western blotting shows the expression change of mitophagy related proteins (LC3, p62 and PARKIN) in CCECs treated with AGES, LIPUS, and/or 3-MA. The expression levels of COX IV were used as a loading reference for total mitochondrial protein. The Western blotting result based on 2 independent repeats with a similar trend. (**B**) Western blotting shows the expression change of corpus cavernosum endothelial markers (CD31 and KIT) in CCECs treated with AGES, LIPUS, and/or 3-MA. The expression levels of ACTB were used as a loading reference for total cell protein. The Western blotting result based on 2 independent repeats with a similar trend. (**C**–**E**) Flow cytometry demonstrates the 3-MA treatment on the early apoptosis (**B**) and late apoptosis (**C**) of CCECs after LIPUS treatment. The existence of Annexin V (*X*-axis) and the nuclear staining of PI (*Y*-axis) by flow cytometry are shown. n = 5 independent repeats. Data are shown as mean ± SD. ns, not significant; *, *p* < 0.05. (**F**,**G**) Phalloidin staining shows typical changes of CCECs tube formation after 3-MA and/or LIPUS treatment. The total mesh area score is used to represent the level of tube formation in each group. n = 5 independent repeats. Data are shown as mean ± SD. ns, not significant; *, *p* < 0.05. AGES: advanced glycation end products; LIPUS: low-intensity pulsed ultrasound; 3-MA: 3-methyladenine.

## Data Availability

RNA-seq data are attached in Appendix A.

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
