# Peer review of "Low-Intensity Pulsed Ultrasound Counteracts Advanced Glycation End Products-Induced Corpus Cavernosal Endothelial Cell Dysfunction via Activating Mitophagy"

_ijms, 2022, doi:10.3390/ijms232314887_

Round 1

Reviewer 1 Report

The manuscript entitled “Low-intensity Pulsed Ultrasound Restores Advanced Glycation End Products-Induced Corpus Cavernosal Endothelial Cells Dysfunction via Activating Mitophagy” addresses the beneficial effects of low-intensity pulsed ultrasound (LIPUS) against corpus cavernosal endothelial cells (CCECs) dysfunction in diabetes mellitus-induced erectile dysfunction (DMED) and the underlying mechanisms. Interestingly, LIPUS counteracted AGES-triggered mitochondria damage and disrupted angiogenesis in CCECs. Meanwhile, LIPUS promoted mitophagy and dampened reactive oxygen species (ROS) production. Notably, the beneficial impact of LIPUS was attenuated when mitophagy was inhibited by 3-methyladenine. The manuscript is clearly written, and the current findings are interesting.

Comments:   

1) In section 4.2. Experimental models and subject details, how did the authors decide on the dose of LIPUS on the CCECs in vitro (LIPUS (50-200 mW/cm2) was stimulated a total of three times, 5 minutes each time with 1-minute intervals)? The authors are advised to address this point and add the answer and proper citation to the comment in section 4.3.

2) Likewise, in the present work, how did the authors decide on the used concentrations of AGES and 3-MA in CCECs? Please, provide proper citations for the used concentrations and the length of incubation time. These citations need to be added for the relevant section in the material and method section.

3) In the current work, the corpus cavernosal endothelial cells (CCECs) were isolated from the tumor margin of penile carcinoma resection. Have the authors considered that the transformed cells of that tumor may not behave exactly as the normal non-transformed CCECs? This may affect the outcomes of the current study. Authors are advised to address this point and add the answer to the discussion section.

4) In line 422, what do the authors specifically mean by “the gradations of blots were detected by chemiluminescence”? Please, clarify.

5) In lines 464-465, the quantification of data for IHC and ICC for each disease type in the study was based on at least two biological repeats. In fact, 2 biological repeats may not give reliable data. Please, address this point and add more repeats.

6) Likewise, in lines 465-466, the quantification of data Scanning electron microscope and transmission electron microscope image (Figure 1A and 1B) was based on at least two biological repeats. In fact, 2 biological repeats may not give reliable data. Please, address this point and add more repeats.

7) In the statistical analysis section, did the authors check data normality and homogeneity before proceeding to one-way ANOVA? Authors are advised to address this point and add the answers to the comment in the material and methods section.

8) To make all figure legends stand-alone, authors are advised to add the full name of the used abbreviations at the end of each legend including LIPUS, CCECs, AGEs, etc.

9) In the figure legends, the authors are advised to add the number of replicates from which data were extracted for all assessed parameters. Authors are advised to address this point and add the answers to the comment to all the relevant figure legends.

10) In the legend of figures 8A and B, the authors are advised to describe the number of replicates used in Western blotting. Moreover, were the data extracted from independent samples?

11) The authors are advised to describe how they scored “total meshes area score” and mitochondria morphology score. Please, add these data to the material and methods section. As implied by the naming scores, are these data discrete (non-continuous)? If yes, the statistics for this type of non-parametric should not be performed using ANOVA analysis test. Instead, for non-parametric data, the authors are advised to analyze the data using Kruskal-Wallis analysis of variance. When statistical significance is obtained, Dunn's test is applied.

12) In the discussion section, authors are advised to describe any reported adverse effects of LIPUS in the clinical setting from the literature.

13) The current title needs to be modified to fit well with the obtained data. The term “restores dysfunction” is confusing to readers as the dysfunction was counteracted or combated by LIPUS. The authors may consider re-writing as “Low-intensity Pulsed Ultrasound Counteracts Advanced Glycation End Products-Induced Corpus Cavernosal Endothelial Cells Dysfunction via Activating Mitophagy”.

Author Response

The manuscript entitled “Low-intensity Pulsed Ultrasound Restores Advanced Glycation End Products-Induced Corpus Cavernosal Endothelial Cells Dysfunction via Activating Mitophagy” addresses the beneficial effects of low-intensity pulsed ultrasound (LIPUS) against corpus cavernosal endothelial cells (CCECs) dysfunction in diabetes mellitus-induced erectile dysfunction (DMED) and the underlying mechanisms. Interestingly, LIPUS counteracted AGES-triggered mitochondria damage and disrupted angiogenesis in CCECs. Meanwhile, LIPUS promoted mitophagy and dampened reactive oxygen species (ROS) production. Notably, the beneficial impact of LIPUS was attenuated when mitophagy was inhibited by 3-methyladenine. The manuscript is clearly written, and the current findings are interesting.

Comments:   

1) In section 4.2. Experimental models and subject detailshow did the authors decide on the dose of LIPUS on the CCECs in vitro (LIPUS (50-200 mW/cm2) was stimulated a total of three times, 5 minutes each time with 1-minute intervals)? The authors are advised to address this point and add the answer and proper citation to the comment in section 4.3.

Response: Thank you for your advice. It is well known that the therapeutic dose of LIPUS depends on its intensity and duration. A clinical study that used energy intensity levels applied to the patients was 300 mW/cm2 [1], and another animal study chose different energy levels (100, 200, and 300 mW /cm2)[2]. Perhaps due to the attenuation effect of the skin, subcutaneous tissue and tunic albumen, the dose used in the in vivo experiment often was high (300 mW/cm2). So we set 300 mW /cm2 as the highest level (Figure 4). Since there was no difference between the 300 mW /cm2 group and the 0 mW /cm2 control group, we set the highest group to 200 in the subsequent experiment (Figure 6 and 7).

As for the treatment duration. For ED patients, the current method of clinic treatment is to use LIPUS to stimulate four parts of the penis for 5 minutes (total of 20 minutes). But different animal and in vitro studies have also chosen different lengths of stimulation duration. Feng-Yi designed an experimental group for 15 min per day [3]. Fang Chen used 100 mW/cm2 to stimulate human microvascular endothelial cells (HMECs) for 10min continuously[4]. In another study, LIPUS activated mitophagy in βcells, the range of duration was from 5 to 15 min, and ultrasound intensity was 100 - 500 mW /cm2 [5]. Theoretically, on the premise of not affecting cell activity, increasing LIPUS load as much as possible could improve the therapeutic effect. Proper interval time may reduce the thermal effect of LIPUS and other adverse effects. Based on the above studies, we chose 5 minutes each time with 1-minute intervals.

We have added some citations about these in the discussion in our revised manuscript.

2) Likewise, in the present work, how did the authors decide on the used concentrations of AGES and 3-MA in CCECs? Please, provide proper citations for the used concentrations and the length of incubation time. These citations need to be added for the relevant section in the material and method section.

Response: Thank you for your question. Previous studies used AGES to stimulate endothelial cells at concentrations was 10-50 ug/ml [6] or 100-300 ug/ml [7]. Combining these results, we used 20 ug/ml for the low-concentration AGE group and 100 ug/ml for the high-concentration AGE group. The concentration of 3-MA inhibiting mitochondrial autophagy was used by 10mM in many studies [8, 9], so we also chose 10mM and treated for 48h.

3) In the current work, the corpus cavernosal endothelial cells (CCECs) were isolated from the tumor margin of penile carcinoma resection. Have the authors considered that the transformed cells of that tumor may not behave exactly as the normal non-transformed CCECs? This may affect the outcomes of the current study. Authors are advised to address this point and add the answer to the discussion section.

Response: It's very difficult to get corpus cavernosum from healthy donors. In this study, all penile tumor patients have normal stimulated erections and early morning erections and are under the age of 50. But your concern is worthy of serious consideration, and we have mentioned this possibility in the discussion section of the revised manuscript. 

4) In line 422, what do the authors specifically mean by “the gradations of blots were detected by chemiluminescence”? Please, clarify.

Response: Sorry for the confusion. “the gradations of blots were detected by chemiluminescence” has been replaced with “the result of western blotting were detected with Gel Imaging System”.

5) In lines 464-465, the quantification of data for IHC and ICC for each disease type in the study was based on at least two biological repeats. In fact, 2 biological repeats may not give reliable data. Please, address this point and add more repeats.

Response: We had 5 repeats in the normal control group. Due to the difficulty in obtaining DMED CC tissue, we only have 2 repeats of DMED CC at present, we have explained this limitation in the discussion section of the revised manuscript. However, the subsequent functional experiments are consistent with what we observed in situ, so this limitation does not affect our main conclusions here.

6) Likewise, in lines 465-466, the quantification of data Scanning electron microscope and transmission electron microscope image (Figure 1A and 1B) was based on at least two biological repeats. In fact, 2 biological repeats may not give reliable data. Please, address this point and add more repeats.

Response: Thank you for your question. Likewise, we also explained this limitation in the discussion section of the revised manuscript.

7) In the statistical analysis section, did the authors check data normality and homogeneity before proceeding to one-way ANOVA? Authors are advised to address this point and add the answers to the comment in the material and methods section.

Response: Thank you for your advice. Due to the small number of repetitions (less than 10) in the experiment, all these data were not normality and homogeneity (checked with the D'Agostino-Pearson and Kolmogorov-Smirnov test in GraphPad v9.0), so we changed all relevant results to Kruskal-Wallis analysis. We have added this description in the material and methods section of the revised manuscript.

8) To make all figure legends stand-alone, authors are advised to add the full name of the used abbreviations at the end of each legend including LIPUS, CCECs, AGEs, etc.

Response: Thank you for your advice. We have added the full name of the used abbreviations at the end of each legend.

9) In the figure legends, the authors are advised to add the number of replicates from which data were extracted for all assessed parameters. Authors are advised to address this point and add the answers to the comment to all the relevant figure legends.

Response: Thank you for your advice. We have added the number of replicates at each figure legend.

10) In the legend of figures 8A and B, the authors are advised to describe the number of replicates used in Western blotting. Moreover, were the data extracted from independent samples?

Response: Thank you for your advice. We have repeated the western blotting base on 3 independent samples and got a similar trend, and we have added this description in the revised manuscript.

11) The authors are advised to describe how they scored “total meshes area score” and mitochondria morphology score. Please, add these data to the material and methods section. As implied by the naming scores, are these data discrete (non-continuous)? If yes, the statistics for this type of non-parametric should not be performed using ANOVA analysis test. Instead, for non-parametric data, the authors are advised to analyze the data using Kruskal-Wallis analysis of variance. When statistical significance is obtained, Dunn's test is applied.

Response: Tube formation analysis including the “total meshes area score” was performed with Angiogenesis Analyze function of ImageJ software[10]. Mitochondrial staining of CCEC was performed using 0.1 ug/ml MitoTracker™ Red CMXRos (M7512, Invitrogen) incubating 30 min, and mitochondrial morphology (fusion and fission) scoring was performed using Mitochondria Analyzer function of ImageJ software. And all these results were analyzed Kruskal-Wallis analysis and Dunn's test in the revised manuscript. Thank you again for your professional advice.

12) In the discussion section, authors are advised to describe any reported adverse effects of LIPUS in the clinical setting from the literature.

Response: Current studies indicate that LIPUS can safely and effectively treat ED patients without significant adverse effects reported [1, 11]. It was also a safe treatment method for other patients such as Neck Pain[12], or low back pain[13]. No adverse events were reported in these studies.

13) The current title needs to be modified to fit well with the obtained data. The term “restores dysfunction” is confusing to readers as the dysfunction was counteracted or combated by LIPUS. The authors may consider re-writing as “Low-intensity Pulsed Ultrasound Counteracts Advanced Glycation End Products-Induced Corpus Cavernosal Endothelial Cells Dysfunction via Activating Mitophagy”.

Response: thank you for your advice, we have revised the title to “Low-intensity Pulsed Ultrasound Counteracts Advanced Glycation End Products-Induced Corpus Cavernosal Endothelial Cells Dysfunction via Activating Mitophagy”

  1. Cui W, Li H, Guan R, Li M, Yang B, Xu Z, et al. Efficacy and safety of novel low-intensity pulsed ultrasound (LIPUS) in treating mild to moderate erectile dysfunction: a multicenter, randomized, double-blind, sham-controlled clinical study. Transl Androl Urol. 2019;8(4):307-19. Epub 2019/09/27. doi: 10.21037/tau.2019.07.03. PubMed PMID: 31555554; PubMed Central PMCID: PMCPMC6732092.
  2. Lei H, Xin H, Guan R, Xu Y, Li H, Tian W, et al. Low-intensity Pulsed Ultrasound Improves Erectile Function in Streptozotocin-induced Type I Diabetic Rats. Urology. 2015;86(6):1241.e11-8. Epub 2015/09/19. doi: 10.1016/j.urology.2015.07.026. PubMed PMID: 26383610.
  3. Chiang PK, Yang FY. A potential treatment of low intensity pulsed ultrasound on cavernous nerve injury for erectile dysfunction. Medical hypotheses. 2019;122:19-21. Epub 2018/12/30. doi: 10.1016/j.mehy.2018.10.014. PubMed PMID: 30593410.
  4. Yu M, Bian Y, Wang L, Chen F. Low-intensity pulsed ultrasound enhances angiogenesis in rabbit capsule tissue that acts as a novel vascular bed in vivo. Advances in clinical and experimental medicine : official organ Wroclaw Medical University. 2021;30(6):581-9. Epub 2021/05/19. doi: 10.17219/acem/134115. PubMed PMID: 34004084.
  5. Guo T, Liu T, Sun Y, Liu X, Xiong R, Li H, et al. Sonodynamic therapy inhibits palmitate-induced beta cell dysfunction via PINK1/Parkin-dependent mitophagy. Cell Death Dis. 2019;10(6):457. Epub 2019/06/13. doi: 10.1038/s41419-019-1695-x. PubMed PMID: 31186419; PubMed Central PMCID: PMCPMC6560035.
  6. Tsai PS, Chiu CY, Sheu ML, Yang CY, Lan KC, Liu SH. Advanced glycation end products activated endothelial-to-mesenchymal transition in pancreatic islet endothelial cells and triggered islet fibrosis in diabetic mice. Chemico-biological interactions. 2021;345:109562. Epub 2021/06/22. doi: 10.1016/j.cbi.2021.109562. PubMed PMID: 34153226.
  7. Deng X, Huang W, Peng J, Zhu TT, Sun XL, Zhou XY, et al. Irisin Alleviates Advanced Glycation End Products-Induced Inflammation and Endothelial Dysfunction via Inhibiting ROS-NLRP3 Inflammasome Signaling. Inflammation. 2018;41(1):260-75. Epub 2017/11/04. doi: 10.1007/s10753-017-0685-3. PubMed PMID: 29098483.
  8. Hirota Y, Yamashita S, Kurihara Y, Jin X, Aihara M, Saigusa T, et al. Mitophagy is primarily due to alternative autophagy and requires the MAPK1 and MAPK14 signaling pathways. Autophagy. 2015;11(2):332-43. Epub 2015/04/02. doi: 10.1080/15548627.2015.1023047. PubMed PMID: 25831013; PubMed Central PMCID: PMCPMC4502654.
  9. Yang L, Yu Y, Kang R, Yang M, Xie M, Wang Z, et al. Up-regulated autophagy by endogenous high mobility group box-1 promotes chemoresistance in leukemia cells. Leukemia & lymphoma. 2012;53(2):315-22. Epub 2011/08/26. doi: 10.3109/10428194.2011.616962. PubMed PMID: 21864037.
  10. Zudaire E, Gambardella L, Kurcz C, Vermeren S. A computational tool for quantitative analysis of vascular networks. PloS one. 2011;6(11):e27385. Epub 2011/11/24. doi: 10.1371/journal.pone.0027385. PubMed PMID: 22110636; PubMed Central PMCID: PMCPMC3217985 SAIC-Frederick, Inc). This does not alter the authors' adherence to all the PLoS ONE policies on sharing data and materials.
  11. Chen H, Li Z, Li X, Yang Y, Dai Y, Xie Z, et al. The Efficacy and Safety of Thrice vs Twice per Week Low-Intensity Pulsed Ultrasound Therapy for Erectile Dysfunction: A Randomized Clinical Trial. J Sex Med. 2022;19(10):1536-45. Epub 2022/08/24. doi: 10.1016/j.jsxm.2022.06.009. PubMed PMID: 35999130.
  12. Qing W, Shi X, Zhang Q, Peng L, He C, Wei Q. Effect of Therapeutic Ultrasound for Neck Pain: A Systematic Review and Meta-Analysis. Archives of physical medicine and rehabilitation. 2021;102(11):2219-30. Epub 2021/03/17. doi: 10.1016/j.apmr.2021.02.009. PubMed PMID: 33722564.
  13. Seco J, Kovacs FM, Urrutia G. The efficacy, safety, effectiveness, and cost-effectiveness of ultrasound and shock wave therapies for low back pain: a systematic review. The spine journal : official journal of the North American Spine Society. 2011;11(10):966-77. Epub 2011/04/13. doi: 10.1016/j.spinee.2011.02.002. PubMed PMID: 21482199.

Reviewer 2 Report

The authors describe the Injury to corpus cavernosal endothelial cells (CCECs) as an important pathological basis of diabetes mellitus-induced erectile dysfunction (DMED). While low-intensity pulsed ultrasound (LIPUS) has been shown to improve erectile function in DMED, its therapeutic mechanism of action remains unclear. The authors used an experimental model of CCECs in the presence of AGES and evaluated the effects of LIPUS and studied mitophagy as a possible mechanism of action.

abstract

Although the abstract is clear, it does not meet the journal's formatting requirements:

1) Background: Place the question addressed in a broad context and highlight the purpose of the study; 2) Methods: Describe briefly the main methods or treatments applied. Include any relevant preregistration numbers, and species and strains of any animals used. 3) Results: Summarize the article's main findings; and 4) Conclusion: Indicate the main conclusions or interpretations. The abstract should be an objective representation of the article: it must not contain results which are not presented and substantiated in the main text and should not exaggerate the main conclusions.

introduction

The introduction is well structured and adequately highlights the importance of the study. The objective is clearly defined, and the hypothesis of the study is highlighted.

materials and methods

- Line 384: How many study subjects (patients) were used?

- Lines 395-396: the authors state that the cells were sorted by MACS with a Dead Cell Removal Kit (130-090-101, Miltenyi Biotec) to remove dead cells. I recommend the authors to detail the MACS sorting procedure performed.

- Lines 407-408: Were the secondary antibodies conjugated to any molecule?

- Lines 420-421: I recommend the authors to make a more detailed description of the antibodies used.

- The authors describe in results section 2.2 and 2.7 the use of flow cytometry, but there is no detailed description of the procedure. It is imperative to describe in detail the use of this technology (equipment used, fluorophores versus filters, compensation details, acquisition speed, etc).

- Lines 459-466: How did the authors measure the normality of the data, what was the measure of dispersion used?

- The authors report having measured early apoptosis and late apoptosis by flow cytometry, but the marker(s) used, and the procedure performed are not described in the material and methods section.

- I suggest the authors consider the following: “Materials and methods should be described with sufficient detail to allow others to replicate and build on published results. New methods and protocols should be described in detail while well-established methods can be briefly described and appropriately cited”

Results

- what does the white arrow in figure 1C indicate?

- Lines 94-96: I recommend the authors to describe in the legend of figure 1D and 1E how the values are expressed (mean+/- standard deviation?), how many times the experiment was repeated?

- I recommend the authors to explain in more detail in each figure legend how the results of the graphs are expressed (percentage, mean, standard deviation, standard error) and the number of replicates of each experiment.

- what is the meaning of NC in the graphs? I recommend authors to describe in the legend all abbreviations used in graphs and figures.

- The legend to figure 4 shows the effects of LIPUS treatment of CCEs under AGES state, but it is not clear whether it is LAGES group or HAGES group.

- Figure 6 shows results of mitosox staining, but this procedure is not described in the material and methods section. I recommend the authors to review all the analyses performed and describe them in the materials and methods section.

- Describe the use of ACTB in materials and methods

Author Response

The authors describe the Injury to corpus cavernosal endothelial cells (CCECs) as an important pathological basis of diabetes mellitus-induced erectile dysfunction (DMED). While low-intensity pulsed ultrasound (LIPUS) has been shown to improve erectile function in DMED, its therapeutic mechanism of action remains unclear. The authors used an experimental model of CCECs in the presence of AGES and evaluated the effects of LIPUS and studied mitophagy as a possible mechanism of action.

abstract

Although the abstract is clear, it does not meet the journal's formatting requirements:

1) Background: Place the question addressed in a broad context and highlight the purpose of the study; 2) Methods: Describe briefly the main methods or treatments applied. Include any relevant preregistration numbers, and species and strains of any animals used. 3) Results: Summarize the article's main findings; and 4) Conclusion: Indicate the main conclusions or interpretations. The abstract should be an objective representation of the article: it must not contain results which are not presented and substantiated in the main text and should not exaggerate the main conclusions.

Response: Thank you for your advice. We have revised our abstract including adding the purpose of the study and the main methods. However, we found that the abstracts of recent articles published in the journal did not use the four-paragraph structure[1, 2], so we stuck with the original format.

introduction

The introduction is well structured and adequately highlights the importance of the study. The objective is clearly defined, and the hypothesis of the study is highlighted.

Response: Thank you for your affirmation.

materials and methods

- Line 384: How many study subjects (patients) were used?

Response: This result based on n = 10 regions from 5 independent samples (penile tumor patients with good stimulated erections and early morning erections). We have added this information in our revised manuscript.

- Lines 395-396: the authors state that the cells were sorted by MACS with a Dead Cell Removal Kit (130-090-101, Miltenyi Biotec) to remove dead cells. I recommend the authors to detail the MACS sorting procedure performed.

Response: 106 cells were co-incubated with 20 ul MicroBeads at 4℃ for 30 min. After blocking and washing, performing magnetic separation with LS Columns (130-042-401; Miltenyi) and MidiMACS Separator (130-042-302; Miltenyi). Retaining cells those were not adsorbed by the column. We have added this description in our revised manuscript. More detail could be found in https://www.miltenyibiotec.com/CN-en/products/dead-cell-removal-kit.html#130-090-101

- Lines 407-408: Were the secondary antibodies conjugated to any molecule?

Response: Alexa Fluor 488 donkey anti-rabbit IgG (A21206, Thermo Fisher), Alexa Fluor 555 donkey anti-rabbit IgG (A31572, Thermo Fisher), Alexa Fluor 488 donkey anti-mouse IgG (A21202, Thermo Fisher) and Alexa Fluor 555 donkey anti-mouse IgG (A31570, Thermo Fisher) were used as secondary antibodies. we have added a more detail information about antibodies in the revised manuscript. 

- Lines 420-421: I recommend the authors to make a more detailed description of the antibodies used.

Response: We have added a more detail information about antibodies in the revised manuscript. 

- The authors describe in results section 2.2 and 2.7 the use of flow cytometry, but there is no detailed description of the procedure. It is imperative to describe in detail the use of this technology (equipment used, fluorophores versus filters, compensation details, acquisition speed, etc).

Response: Annexin V-FITC apoptosis detection Kit (A211-01, Vazyme Biotech) was used to measure cell apoptosis according to the manufacturer’s protocol. In brief, CCECs were collected by EDTA-free trypsin, washed with ice-cold phosphate-buffered saline twice, and resuspended in 100ul binding buffer. Next, the cells were incubated with 5ul An-nexin V-FITC and 5ul PI staining solution in the dark for 10 min, and then diluted with binding buffer to 500ul. Cell apoptosis was subsequently analyzed by a flow cytometry (Beckman Coulter, Inc. Brea, CA, USA), the fluorescence signals of Annexin V and PI conjugate were detected in fluorescence intensity channels B525-FITC and Y585-PE without compensation set, respectively. Acquisition speed was 60ul/min.The apoptotic rates were measured by FlowJo software, version 10.6.1 (FlowJo LCC, Becton Dickinson, Ashland, OR, USA). We have added this information in revised manuscript.

- Lines 459-466: How did the authors measure the normality of the data, what was the measure of dispersion used?

Response: Data were check normality with the D'Agostino-Pearson and Kolmogorov-Smirnov test by GraphPad Prism 9.0.0 software. And according to another reviewer’s advice, we have change our statistics analysis to Kruskal-Wallis analysis and Dunn's test.

- The authors report having measured early apoptosis and late apoptosis by flow cytometry, but the marker(s) used, and the procedure performed are not described in the material and methods section.

Response: We have added this information in revised manuscript.

- I suggest the authors consider the following: “Materials and methods should be described with sufficient detail to allow others to replicate and build on published results. New methods and protocols should be described in detail while well-established methods can be briefly described and appropriately cited”

Response: Thank you for your advice. We have added relate description and citation in revised manuscript.

Results

- what does the white arrow in figure 1C indicate?

Response: The white arrow points to γH2AX and VWF double positive CCECs, and VWF-negative non-endothelial cells are marked in the dotted box. We have added this description in the legend of figure 1C in revised manuscript.

- Lines 94-96: I recommend the authors to describe in the legend of figure 1D and 1E how the values are expressed (mean+/- standard deviation?), how many times the experiment was repeated?

Response: It is a ratio between γH2AX+ and γH2AX-. Data was collected according to 300 cells from 5 Normal CC samples and 300 cells 2 DMED CC samples. We have added this description in the legend of revised manuscript.

- I recommend the authors to explain in more detail in each figure legend how the results of the graphs are expressed (percentage, mean, standard deviation, standard error) and the number of replicates of each experiment.

Response: We have added this information in each legend of revised manuscript.

- what is the meaning of NC in the graphs? I recommend authors to describe in the legend all abbreviations used in graphs and figures.

Response: We have added this information in each legend of revised manuscript.

- The legend to figure 4 shows the effects of LIPUS treatment of CCEs under AGES state, but it is not clear whether it is LAGES group or HAGES group.

Response: Sorry for that confusion. It is HAGES group. We have added this information in the figure and its legend of revised manuscript.

- Figure 6 shows results of mitosox staining, but this procedure is not described in the material and methods section. I recommend the authors to review all the analyses performed and describe them in the materials and methods section.

Response: Sorry for that information missing. Mitochondrial ROS staining of CCEC was performed using 0.5 ug/ml MitoTracker™ Red CMXRos (M36007, Invitrogen). The cytoskeleton staining of CCEC was performed using 1:500 diluted FITC-Phalloidin (G1028, Servicebio). Mitochondrial staining of CCEC was performed using 0.1 ug/ml MitoTracker™ Red CMXRos (M7512, Invitrogen) incu-bating 30 min, and mitochondrial morphology (fusion and fission) scoring was per-formed using Mitochondria Analyzer function of ImageJ software. We have added all this information in the materials and methods section of revised manuscript.

- Describe the use of ACTB in materials and methods

Response: We have added this information in the materials and methods section of revised manuscript.

  1. Pandey S, Madreiter-Sokolowski CT, Mangmool S, Parichatikanond W. High Glucose-Induced Cardiomyocyte Damage Involves Interplay between Endothelin ET-1/ETA/ETB Receptor and mTOR Pathway. 2022;23(22):13816. PubMed PMID: doi:10.3390/ijms232213816.
  2. Kitagawa S, Tang C, Unekawa M, Kayama Y, Nakahara J, Shibata M. Sustained Effects of CGRP Blockade on Cortical Spreading Depolarization-Induced Alterations in Facial Heat Pain Threshold, Light Aversiveness, and Locomotive Activity in the Light Environment. 2022;23(22):13807. PubMed PMID: doi:10.3390/ijms232213807.
